# PSI Spatially Constrained Clustering: The Sibari and Metaponto Coastal Plains

**Nicola Amoroso** [1,2], **Roberto Cilli** [3,*], **Davide Oscar Nitti** [4], **Raffaele Nutricato** [4], **Muzaffer Can Iban** [5], **Tommaso Maggipinto** [2,3], **Sabina Tangaro** [2,6], **Alfonso Monaco** [2,3,†] **and Roberto Bellotti** [2,3,†]

1   Dipartimento di Farmacia-Scienze del Farmaco, Università degli Studi di Bari Aldo Moro, 70125 Bari, Italy; nicola.amoroso@uniba.it
2   Istituto Nazionale di Fisica Nucleare, Sezione di Bari, 70125 Bari, Italy; tommaso.maggipinto@uniba.it (T.M.); sabina.tangaro@uniba.it (S.T.); alfonso.monaco@uniba.it (A.M.)
3   Dipartimento Interateneo di Fisica, Università degli Studi di Bari Aldo Moro, 70125 Bari, Italy
4   Geophysical Applications Processing—GAP s.r.l, 70125 Bari, Italy; davide.nitti@gapsrl.eu (D.O.N.); raffaele.nutricato@gapsrl.eu (R.N.)
5   Department of Geomatics Engineering, Çiftlikköy Campus, Mersin University, 33343 Mersin, Türkiye; caniban@mersin.edu.tr
6   Dipartimento di Scienze del Suolo, della Pianta e degli Alimenti, Università degli Studi di Bari Aldo Moro, 70125 Bari, Italy
*   Correspondence: roberto.cilli@uniba.it
†   These authors contributed equally to this work.

**Abstract:** PSI data are extremely useful for monitoring on-ground displacements. In many cases, clustering algorithms are adopted to highlight the presence of homogeneous patterns; however, clustering algorithms can fail to consider spatial constraints and be poorly specific in revealing patterns at lower scales or possible anomalies. Hence, we proposed a novel framework which combines a spatially-constrained clustering algorithm (SKATER) with a hypothesis testing procedure which evaluates and establishes the presence of significant local spatial correlations, namely the LISA method. The designed workflow ensures the retrieval of homogeneous clusters and a reliable anomaly detection; to validate this workflow, we collected Sentinel-1 time series from the Sibari and Metaponto coastal plains in Italy, ranging from 2015 to 2021. This particular study area is interesting due to the presence of important industrial and agricultural settlements. The proposed workflow effectively outlines the presence of both subsidence and uplifting that deserve to be focused and continuous monitoring, both for environmental and infrastructural purposes.

**Keywords:** environmental monitoring; ground displacements; persistent scatterers; SKATER; LISA

## 1. Introduction

Since its foundations, Persistent Scatter Interferometry (PSI) has shown great potential for several applications [1,2]; in particular, its contribution to monitoring geophysical phenomena such as subsidence and uplift (driven by environmental forces or human activities) is of paramount importance [3]. The advantages of PSI are manifest as, just to mention a few, it allows fast and easy access to the observation of wide areas and provides measurements with high spatial density based on satellite-borne Synthetic Aperture Radar (SAR). Accordingly, in recent years, a consistent number of studies have proposed and investigated its use. In particular, studies addressing urban subsidence [4–7], mine subsidence [8,9], industrial-related processes [10–12], and coastal monitoring [13–16] can be mentioned.

PSI relies on a single working principle, the presence of stable reflectors, i.e., persistent scatterers, which can be used to achieve highly accurate differential measurements [17]. Several different techniques have been proposed [18–22]. In particular, PSI techniques are extremely helpful when dealing with slow-occurring phenomena such as subsidence, tectonic uplifts, and ground deformation processes in civil engineering structures [23].

Here, the SPINUA (Stable Point Interferometry over Unurbanized Areas) algorithm [24] was used to process Sentinel-1 data and highlight occurring displacements along the line of sight.

The main goal of PSI analyses is providing displacement maps which can be suitably used to identify ground displacements. However, in many cases, further evaluations are needed to identify the presence of anomalous patterns or outlier phenomena. A common choice is to use clustering algorithms [25–28], whose underlying assumption (widely accepted by the scientific community) is that the more PS show a coherent displacement, the more reliable the observed effect is. A popular choice for the remote sensing community is the DBSCAN algorithm (Density-Based Spatial Clustering of Applications with Noise) [29–31], especially for its efficiency in retrieving clusters with arbitrary shape and its computational efficiency. Nevertheless, as DBSCAN operates in the feature space, it can neglect important constraints provided by spatial proximity, which can, in principle, improve clustering results. Hence, other strategies, which, directly or indirectly, take into account spatial proximity have been proposed [32–35]. Among them, we proposed the adoption of the SKATER clustering algorithm (Spatial 'K'luster Analysis by Tree Edge Removal) [36] for two main reasons: (i) SKATER is easy to tune, as it fundamentally depends only on one hyper-parameter, the number of classes, and (ii) it is computationally efficient. In fact, it is based on recursive partitioning of a minimal spanning tree, which transforms an np-hard problem in a quasi-linear one [37]; this allows the processing of data of medium–large sample size, including $\sim 10^5$ observations, faster than other algorithms [38].

However, given the wide heterogeneity of the phenomena affecting the ground surface and the already mentioned high variability of displacements, it is not uncommon to observe clusters that are poorly specific, often grouping together pixels which should be considered apart. Of course, this issue is a direct consequence of clustering inherent "ill-posedness" [39]. Nevertheless, remote sensing applications have an advantage, in that spatial proximity is not only a constraint which can be useful to support clusters' partition, but it can be also useful to identify anomalous behaviors. Accordingly, we proposed a procedure which combines the SKATER clustering with a following analysis of spatial association based on the Moran's index, namely the Local Indicators of Spatial Association (LISA) algorithm [40]. Thus, statistics based on spatial proximity were embedded in a processing pipeline to ensure clusters' homogeneity at all scales and highlight the presence of possible anomalies.

The aim of this work was to demonstrate that a procedure combining both the SKATER and LISA algorithms can effectively detect relevant surface phenomena that may need further investigations when performing exploratory analyses on a regional scale. To test and validate this pipeline, we considered the coastal plains of a region in Southern Italy, namely the Sibari and Metaponto plains, which have already been studied in the recent past, for the occurrence of several features of interest, such as the presence of important industrial and agricultural infrastructures, archaeological remains of ancient Magna Graecia settlements, and a not-trivial geological environment including alluvial fans and several marine terraces [41,42]. Additionally, the presence of significant anthropogenic pressure [43] and possible interactions of subsidence with seismic or tectonic activity [44–47] make the continuous monitoring of this region extremely challenging and interesting.

## 2. Mapping the Sibari and Metaponto Coastal Plains

### 2.1. Geography of the Region of Interest

In this work, we considered an area of interest including the Sibari and the Metaponto coastal plains; this area is located in Southern Italy across the Basilicata and Calabria regions. In particular, we focused on the central-northern part of the Sibari plain, including the coastal areas Sibari and Trebisacce-Villapiana, and the southern portion of the Metaponto plain, including the coastal area of Policoro. The region is located in the northern Calabrian arc. It extends for 500 square kilometers and it is confined to the west by the Calabrian Apennines, to the north and to the south by the Pollino and Sila massifs, respectively; finally, the region is delimited to the east by the Ionian Sea.

Subsidence plains are mainly caused by sediment compaction under the pressure of overlying sediments; this can also be worsened by anthropogenic pressure on the seaside localities and groundwater withdrawal in the industrial and urban areas [48]. The region is crossed by multiple rivers which contribute to increasing the hydro-geological risk of the area and expose the area to floods, although embankments have considerably reduced this risk [49]. The region also includes capable faults which were identified and georeferenced by the ITHACA project (ITaly HAzards from CApable faulting) [50].

Concerning the Sibari plain, in the northern sector, the main geomorphological elements are the alluvial fans of Raganello River, Satanasso Fiumara, and Saraceno Fiumara. The Metaponto floodplain, located east of the Bradanic Trough, is mainly derived from the expansions of several rivers: Basento, Bradano, Agri, Sinni, and Cavone; it is a wide sedimentary basin of Plio-Pleistocene followed by Holocene and recent alluvial deposits [51,52]. The elements of interest along with the tectonic setting of the area, capable faults and subduction lines, are reported in Figure 1.

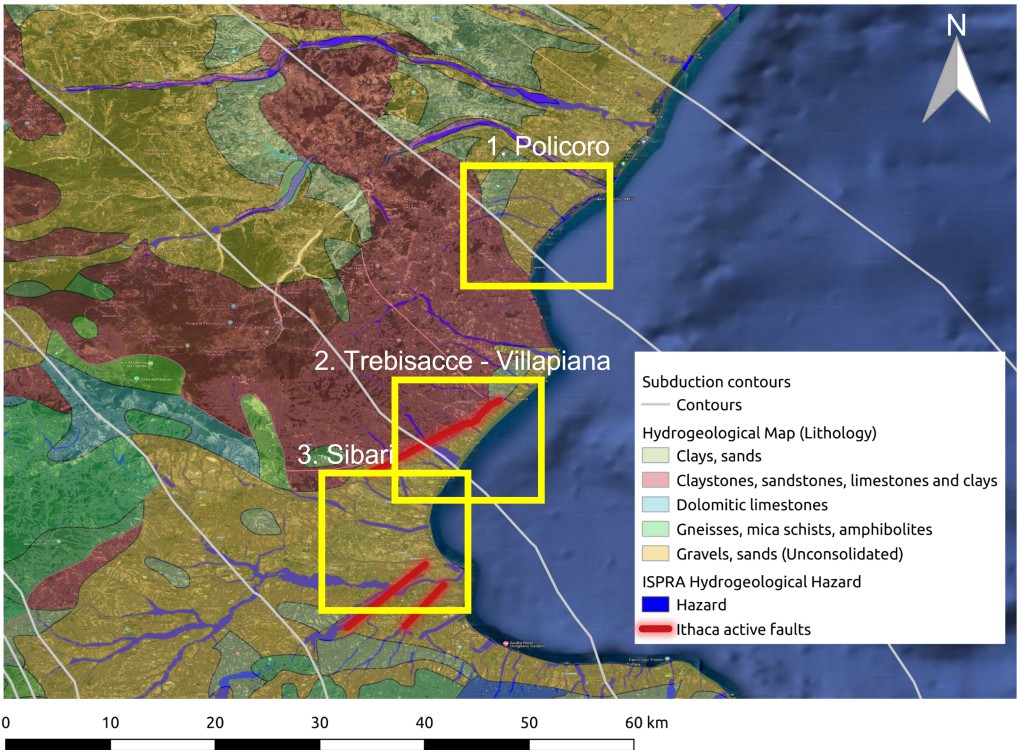

**Figure 1.** Map of the lithological units, active faults, and subduction contours of the areas of concerns.

In the north, the Lauropoli-Trebisacce fault in the SW–NE direction (visible on the map between Villapiana and Trebisacce) is worth mentioning. According to ITHACA, the active faults of Crati (along the river Crati) and of Timparelle, which continue in the SW–NE direction crossing the archaeological area of the old Sybaris, can also be observed. Despite this consolidated knowledge of the area of interest, it is worth noting how some elements are still debated, such as for example the contributions of the faults and subduction lines to the evolution of the Sibari coastal plain [53]. The Metaponto plain presents an interesting diversity in terms of geological elements; four distinct regions can be recognized: Subappennine Clays, marine terraces, alluvial deposits, and the actual coastal region. This peculiar morphology makes the region particularly subjected to seawater intrusion risks [54]. Hence, a continuous monitoring of the region can play a relevant role for both environmental monitoring and management purposes.

## 2.2. The SPINUA Algorithm for Ground Displacement Evaluations

We used Sentinel-1 C-band images (central frequency 5.4 GHz and wavelength 5.6 cm). The Sentinel-1 constellation were composed of two twin satellites (Sentinel-1A and Sentinel-1B, respectively); the first one has been active from October 2014 while the second one stopped its activity in December 2021 after a permanent failure of the Sentinel-1B payload. The two satellites observe the Earth from an altitude of about 693 km, at a nominal ground resolution of about $5 \times 20$ m$^2$ (range $\times$ azimuth) and with a revisit time of 6 days at the equator. The study area is covered along three satellite tracks; for this study the ascending geometry was used. The properties of the data sets of collected ground displacements are outlined in Table 1.

**Table 1.** PSInSAR datasets used for the present study.

| ROI | Orbit | No. of Images | No. of PSs | Time Span |
|---|---|---|---|---|
| Sibari | Asc | 248 | 38,386 | 2 January 2017 to 22 February 2021 |
| Trebisacce-Villapiana | Asc | 190 | 24,574 | 1 April 2015 to 15 February 2019 |
| Policoro | Asc | 204 | 38,265 | 1 April 2015 to 5 March 2019 |

Each dataset consisted of a number of $2.0 \sim 4.0 \times 10^4$ persistent scatterers. We used the SPINUA processing chain to evaluate terrain displacements. For each PS, additional information about height, latitude and longitude, coherence, head angle, and incident angle were also available. A fundamental issue for PSI analyses concerns data coherence. In fact, as ground movements are derived by phase-shift differences, incoherent measures can yield noisy and unreliable results; accordingly, for the present analyses, we selected the time series whose phase coherence exceeded the 0.7 threshold value [55], which ensures in this case a root mean square error (RMSE) below 4 mm for each displacement measurement. Additionally, we removed from the analyses the points laying in uninhabited areas or exceeding the altitude of 50 m, which exceeded the coastal plains. Hence, approximately 50% of the time series were held for subsequent analyses.

Finally, we computed the average velocity along the line-of-sight (LOS) of the remaining observations. These LOS velocities along with the coordinates of the related PSs were used to characterize ground displacements within the region of interest, identify specific homogeneous patterns (such as those caused by subsidence phenomena, debris flows along alluvial fans or seismically-induced uplifts), and provide an overall monitoring service of the region.

## 3. Assessment of Homogeneous and Anomalous Ground Displacements

### 3.1. Methodological Overview

In this work, we presented a workflow to enforce the identification of homogeneous PSI clusters and highlighted the presence, within these clusters, of local patterns and possible anomalies; to this aim, we designed a two-step procedure based on the spatially constrained clustering algorithm SKATER and the outlier/hotspot detection performed by LISA. A schematic overview is presented in Figure 2.

PSI data were used to reconstruct time series of on-ground displacements; these data were then used to feed the SKATER clustering. SKATER exploits spatial constraints to retrieve homogeneous clusters; nevertheless, some clusters can include local patterns which could deserve an independent description or anomalies can remain concealed and, in any case, a statistical assessment of the retrieved clusters is needed; therefore, the LISA method was finally adopted to evaluate the clusters' spatial coherence and highlight the presence of possible anomalies. The SKATER and LISA methods are available in the R package *rgeoda v0.0.10-2* [56].

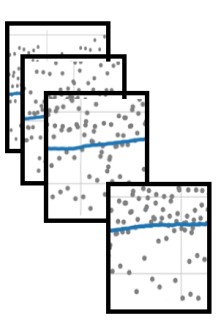

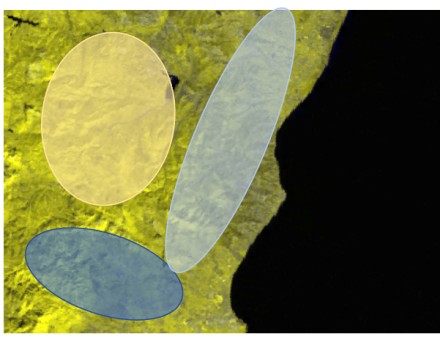

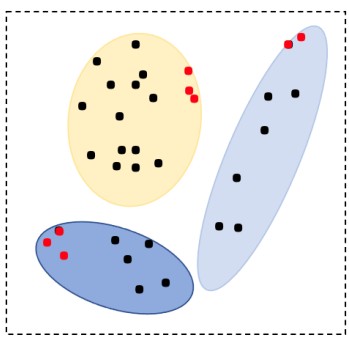

(a) PSI time-series stack      (b) SKATER − spatially constrained clusters      (c) LISA − anomaly detection

**Figure 2.** PSI analyses are carried out to reconstruct time series of on-ground displacements (**a**); time series undergo then the SKATER spatially constrained cluster analysis (**b**); finally, the LISA method is considered to highlight within each clusters coherent local patterns or possible anomalies as depicted in red (**c**).

### 3.2. Spatially-Constrained Clustering Algorithm (SKATER)

One of the main aspects of the present work was the adoption of a spatially-constrained clustering algorithm, namely the SKATER algorithm, in order to group PSs related to the same phenomena by taking into account their spatial proximity. SKATER's basic idea consists in measuring the pairwise distances between all available PS locations so that a symmetric matrix of distances is obtained: in graph theory, this matrix is usually called an adjacency matrix and it can be used to define a connectivity graph. Let $\mathcal{N}$ be the set of PS locations, also called nodes of the graph, then the weighted adjacency matrix element $w_{ij}$ represents the proximity between node $i$ and $j$ (usually the distance reciprocal or a normalized version are considered). The matrix is symmetric as, of course, $w_{ij} = w_{ji}$.

Once the graph is defined, a minimum spanning tree (MST) can be determined. By definition, an MST is a subset of the edges of the original graph allowing to reach all the nodes, i.e., PSs in this specific case, with a minimum number of edges. Accordingly, in this representation, there are no isolated nodes and if further edges are removed, two or more sub-graphs or sub-trees $T_i$ are obtained. These sub-trees can be naturally adopted to reveal spatial clusters. Of course, removing different edges leads to different partitions of the graph. The SKATER algorithm searches for the set of links that, if pruned, generates a partition of sub-trees as homogeneous as possible. For each partition $\Pi = T_1, \ldots, T_K$, the homogeneity is measured by minimizing the sum of the intracluster square deviations $Q(\Pi)$:

$$Q(\Pi) = \sum_{i=1}^{K} SDD_i = \sum_{i=1}^{K} \left( \sum_{j=1}^{N_i} (v_j - \bar{v})^2 \right), \tag{1}$$

where $K$ is the cardinality of the partition $\Pi$ and $SDD_i$ is referred to as the intracluster sum of square deviations computed for the sub-tree $T_i$.

According to this procedure, the main parameter on which SKATER relies is the cardinality $K$, i.e., the number of desired clusters. In fact, if $K$ clusters are desired, $K - 1$ edges must be removed. Initially, all nodes belong to a single class: removing the first edge yields two sub-trees, then removing another edge separates one of these sub-trees in two; the procedure can be iterated until the number of desired spatial clusters is obtained. The edges to remove are those maximizing the partition homogeneity.

It is worth noting that the exhaustive investigation of all possible partitions easily involve extreme computational burdens. This is why SKATER adopts an heuristic approach for fast tree pruning. For each sub-tree, a central node $V_c$ is defined and, then, the cost function $C$ related to cutting each sub-tree starting from the links that connects $V_c$ to its neighbours is computed. Finally, SKATER searches for the optimum cut of each sub-tree in the direction in which $C$ increases and the search ends when the best possible solution is achieved.

Regarding the optimal number clusters $K^*$, the ratio between the between-clusters sum-of-squares $BSS$ and the total-sum-of-squares $TSS$ is computed for each partition obtained by varying the number $K$ of clusters. While $BSS$ measures the squared average distance between all centroids, $TSS$ evaluates the average distance of all points from their overall Euclidean mean; accordingly, their ratio is a measure of clusters' dispersion ranging from 0 (complete overlap of clusters—worst scenario) to 1 (perfect separation—best scenario). For each dataset, we found the optimal number of cluster $K^*$ with the elbow method, i.e., by visually inspecting the $BSS/TSS$ versus $K$ plot and, therefore, selecting the $K^*$ value for which, when increasing the number of clusters, no significant improvement in the overall quality was observed.

*3.3. LISA Outlier Detection*

Within a cluster, it is not uncommon to find smaller regions, even composed of few observations, which seem to not be homogeneous with the surroundings. The reasons are manifold. For example, especially when considering large clusters, the large dimensions can conceal phenomena occurring at lower scales; another confounding situation can occur at clusters' borders where it is probable that points accounting for different phenomena (e.g, moving upwards and downwards) can be spatially close. Hence, we adopted the LISA method to examine the Moran's statistics for spatial auto-correlation of the LOS velocities measured through PSI. Moran's statistics exploits the adjacency matrix $w_{ij}$ previously defined; first of all, the matrix is binarized so that matrix elements are set to 0 if their distance exceeds a threshold, 1 otherwise. In the present study, we set a distance threshold of 30 meters in order to obtain a sparse adjacency matrix. Sparsity is in fact an essential condition in order to decrease the computational burden and, more importantly, to relate this measure with a spatially limited region.

Each adjacency matrix element $w_{ij}$ is related to a PS with $(x_i, y_i)$ coordinates and velocity $v_i$; considering its surroundings, it is possible to introduce the the spatial lagged velocity $v_{i,lag}$:

$$v_{lag} = \frac{\sum_j w_{i,j} v_j}{\sum_j w_{i,j}},$$

(2)

which can be interpreted as the weighted average of LOS velocity of the neighbouring points of the $i^{th}$ observation. According to this definition, the $v_{i,lag}$ of an $(x_i, y_i)$ point depends on the number of considered neighbor points; thus, the sparsity condition ensures that the sum includes few terms.

Examining the scatter plot of the actual velocities and the lagged ones, important considerations can arise; for example, if velocities are described by homogeneous patterns, $v_{lag}$ and $v$ must align and the slope of the straight line should be close to one; points that lie far from this line are spatial outliers. Additionally, the straight line extremities define the so-called "coldspots", at low $v$ and $v_{lag}$ values and "hotspots", at high $v$ and $v_{lag}$ values, of ground velocities. These points correspond to spatial associations of, respectively, low and high values of LOS average velocities. Thanks to the Moran's index $I$, a quantitative evaluation can be carried out by means of a hypothesis test. The index $I$ is the analogous of the Pearson's correlation in spatial terms and it is defined as follows:

$$I = \frac{1}{\sum_{i=1}^{N} \sum_{j=1}^{N} w_{i,j}} \frac{\sum_{i=1}^{N} \sum_{j=1}^{N} w_{i,j}(v_i - \bar{v})(v_j - \bar{v})}{\sum_{l=1}^{N}(v_l - \bar{v})^2},$$

(3)

where $N$ and $\bar{v}$ indicate, respectively, the total number of spatial observations and their average velocity. The index $I$ ranges from $-1$ and $+1$, with $+1$ representing the maximum spatial correlation and $-1$ anti-correlation: in the first case, the neighbor points are perfectly homogeneous and can be clusterized; in the second case, each point is different from its neighbors.

The index $I$ provides a global spatial statistics, which can suitably outline the presence of spatial patterns or anomalies. The LISA approach effectively outlines and localizes these situations by adopting the local Moran's index $I_i$ of the $i^{th}$ observation:

$$I_i = (v_i - \bar{v}) \frac{N}{\sum_{i=1}^{N} \sum_{j=1}^{N} w_{i,j}} \frac{\sum_{j=1}^{N} w_{i,j}(v_j - \bar{v})}{\sum_{l=1}^{N} (v_l - \bar{v})^2}, \tag{4}$$

with the $N$ numerator ensuring that $< I_i >= I$.

After computing the local Moran's indexes $I_i$, the hypothesis testing can be performed to determine whether spatial (anti-)correlations occur. The testing is performed by comparing the experimental values $I_i$ with the Moran's index values expected with a random spatial distribution. In particular, the PSs whose local Moran's index exceeds the average by two standard deviations are considered homogeneous and belonging to the same cluster while the others are spatial outliers.

## 4. Results

### 4.1. Revealing Homogeneous On-Ground Displacements with SKATER

First of all, we examined the presence of homogeneous patterns in the regions of interest by varying the number of expected clusters and computing the corresponding BSS/TSS metrics. By visual inspection, considerations based on the elbow method suggest, for each region, that the optimal number of classes is two or three, see Figure 3.

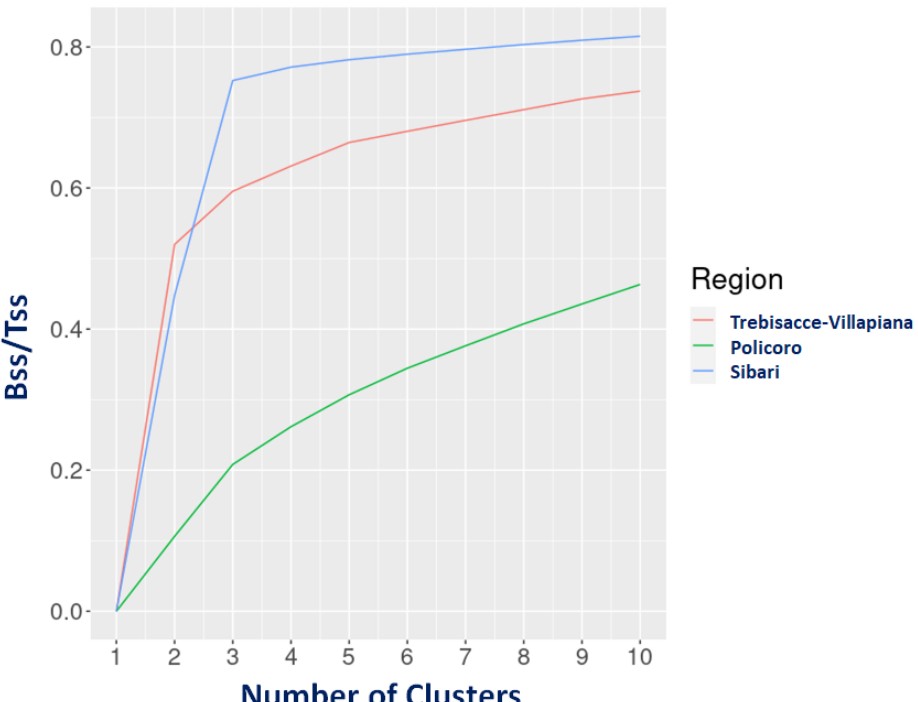

**Figure 3.** Plots comparing the quality of the partition against the number of clusters in terms of the BSS/TSS ratio.

The BSS/TSS ratio shows, manifestly, two distinct phases: a first steepen increase is followed by a much slower incremental behavior (more evident for Sibari and Trebisacce-Villapiana). The number of spatially constrained communities is two for Trebisacce-Villapiana and three for Sibari and Policoro. The areas of Sibari and Trebisacce-Villapiana show a good quality clustering in terms of the BSS/TSS ratio, which reaches values ∼0.8. Conversely, the clustering obtained for the Policoro area seems to be unreliable (BSS/TSS ∼0.4). The partitions returned by SKATER for Trebisacce-Villapiana and Policoro, with the three optimal clusters, are shown in Figure 4.

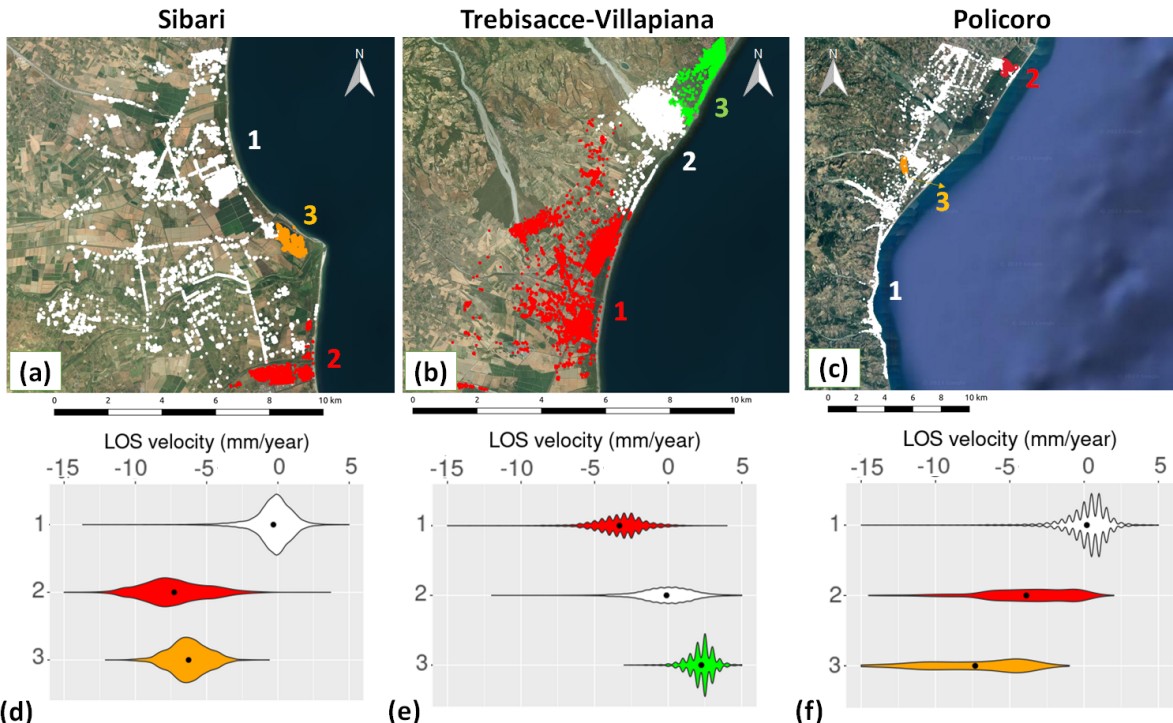

**Figure 4.** On the top: SKATER optimal clustering for Sibari (**a**), Trebisacce-Villapiana (**b**) and Policoro area (**c**); the violin plots on the bottom show the velocity distributions of each optimal cluster. The color code links each spatial cluster to its velocity distribution (**d**–**f**).

Both Sibari and Trebisacce-Villapiana coastal plains are best characterized by three clusters; violin plots allow to appreciate how stable are the clusters, in that LOS velocities appear in general closely distributed to the average values. Nevertheless, more extreme values are present as shown by the violins' long tails. This result suggests the need for a further and localized inspection of the SKATER clusters. Analogously, Policoro can be separated in three clusters whose velocities are well separated, but the overall clustering quality remains poor because of the limited size of the observed clusters, related only to a bridge and a small fraction of Policoro.

Finally, for validating the clustering results by visual inspection, Figure 5 shows the velocity distributions in the region of interest as retrieved by SPINUA.

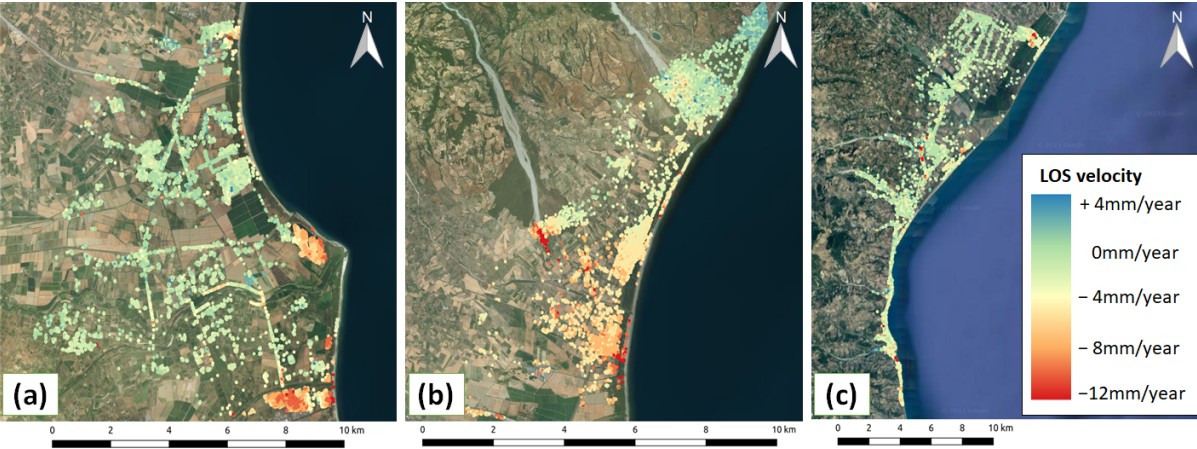

**Figure 5.** SPINUA measured velocities for Sibari (**a**), Trebisacce-Villapiana (**b**) and Policoro area (**c**).

It is worth noting that, choosing suitable colour maps and velocity ranges, the velocity LOS distributions emphasise the presence of three clusters both in Sibari and Trebisacce-Villapiana coastal plains (as suggested by SKATER), while Policoro clusterization remains elusive. Further details about the specific patterns retrieved within each region of interest will be provided in the following sections.

### 4.2. Sibari

To highlight the presence of local patterns or anomalies within the SKATER clusterization of Sibari, further analyses were carried by means of the LISA approach. Figure 6 shows LISA results for this region.

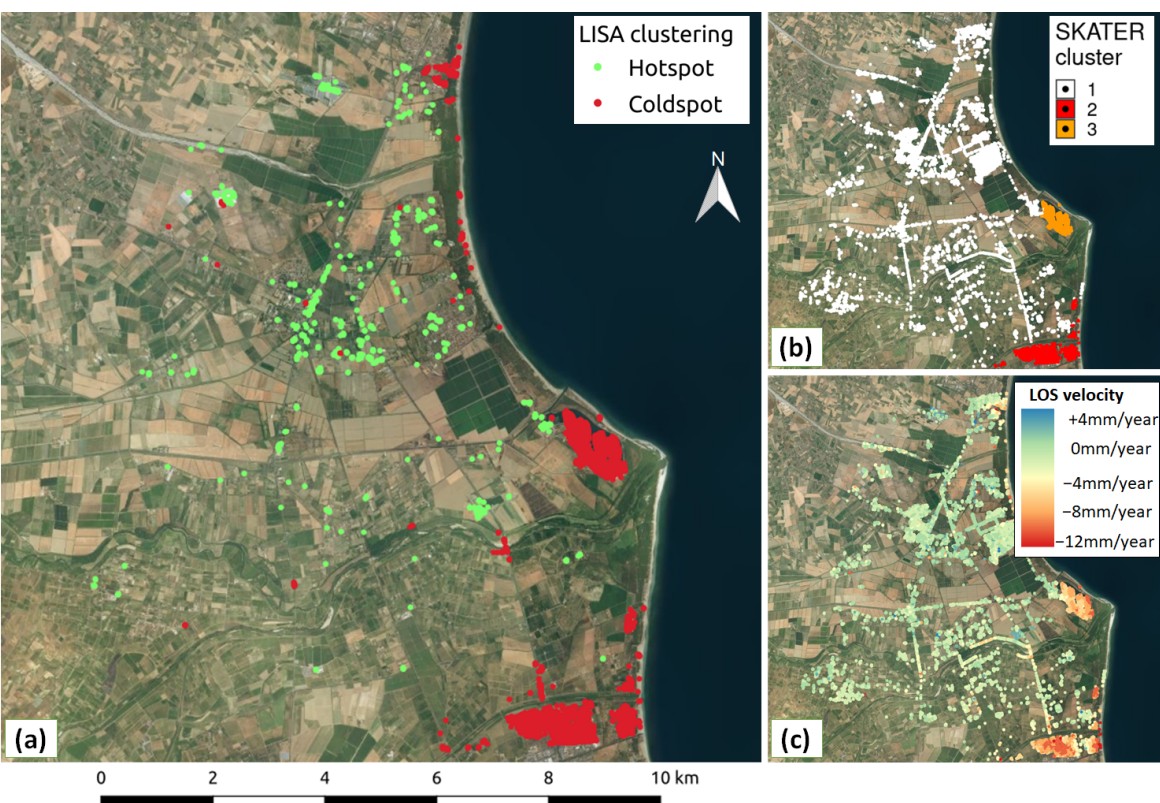

**Figure 6.** LISA analyses of Sibari: coldspots (red) and hotspots (green) are shown (**a**). SKATER optimal clustering is shown in panel (**b**); the spatial distribution of the LOS velocities retrieved by the SPINUA algorithm is shown in panel (**c**).

Green points are related to areas where a significant spatial aggregation of positive LOS velocity occurs. Red points are used in the same situation but with negative LOS velocities. Finally, points not exhibiting a significant spatial auto-correlation are white. Within the Sibari region, while the vast majority of points were stable, some interesting coldspots were also present, for example near Corigliano Calabro and the Sibari lakes area, see Figure 7.

Interestingly, concerning, Corigliano Calabro, the subsidence region is located within its industrial area; it is worth noting that the geometric center of this coldspot corresponds to the known coordinates of a water well. Additionally, a few kilometers towards the coast, it is possible to detect another community of coherent subsidence, corresponding to the Selicetti fraction; in particular, this subsidence ($1 \sim 2$ cm per year) occurs near the coast where several resorts are present. Another interesting subsidence area (1 cm per year) is the one located around the Sibari lakes. This area hosts several residential complexes and an important port.

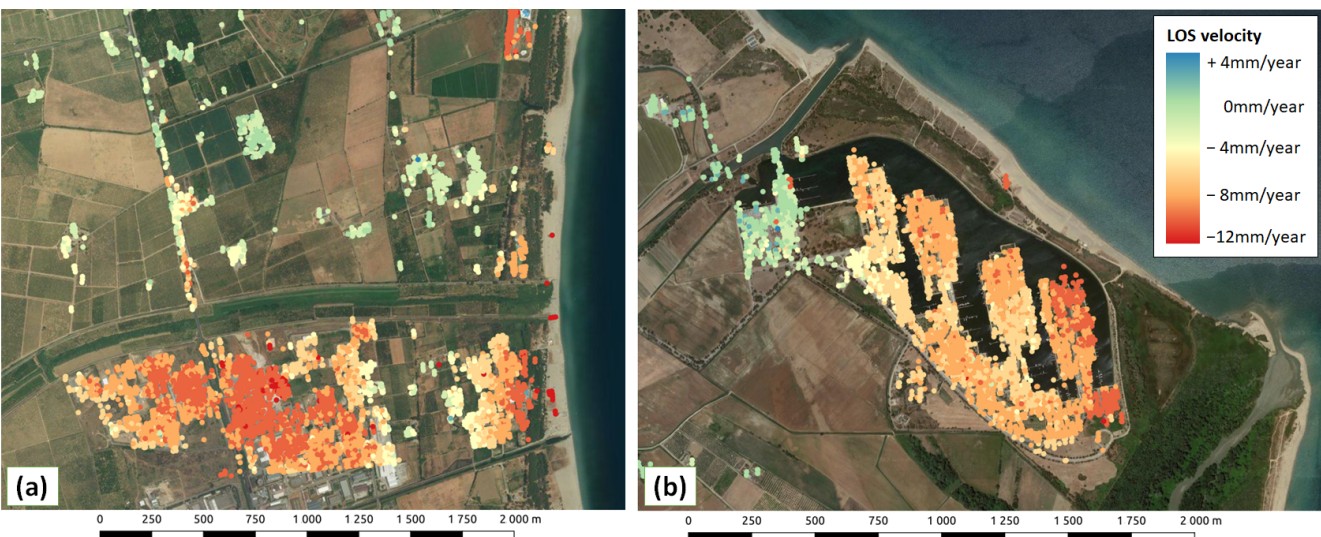

**Figure 7.** The industrial area of Corigliano Calabro (**a**) and the Sibari lakes (**b**) are shown. These areas are two examples of coldspots in the Sibari region.

### 4.3. Trebisacce-Villapiana

We considered the clusterization of Trebisacce-Villapiana and, even in this case, we investigated the presence of possible sub-clusters or patterns missed by SKATER. Results are presented in Figure 8.

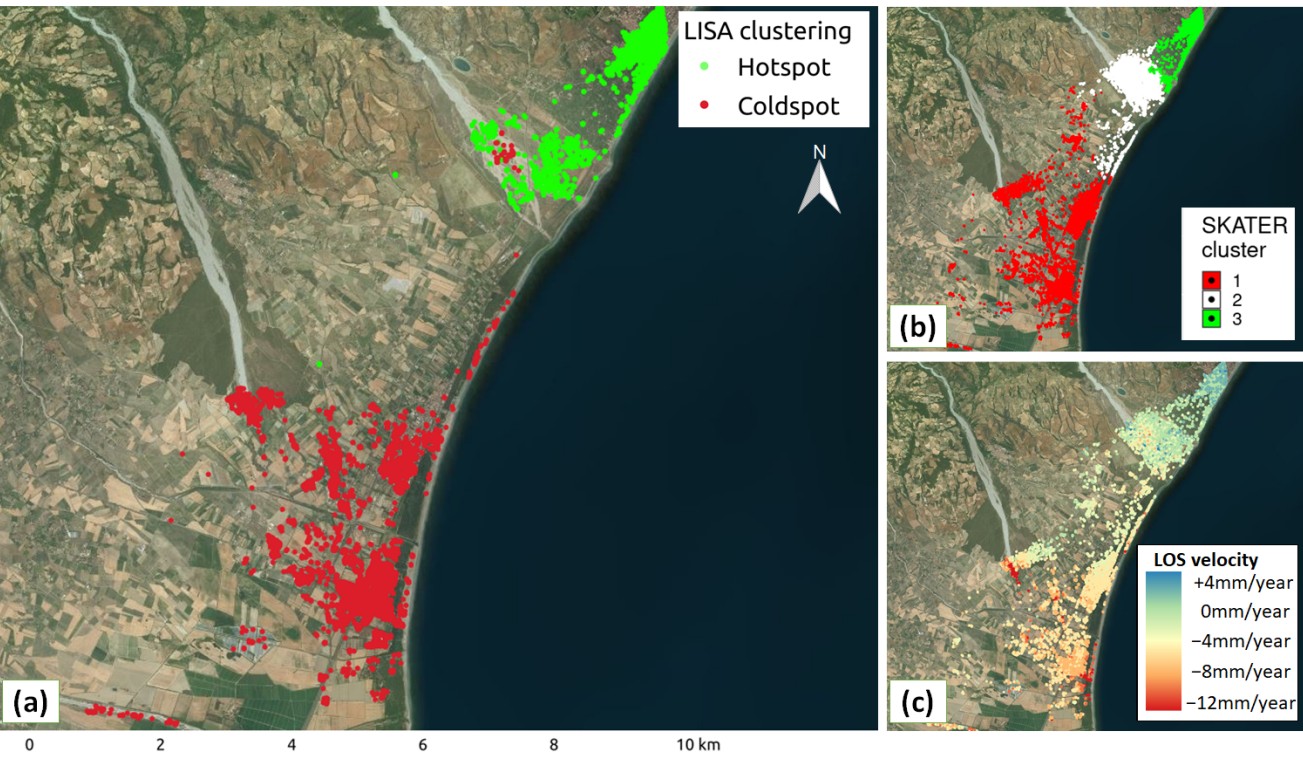

**Figure 8.** LISA analyses of Trebisacce-Villapiana: coldspots (red) and hotspots (green) are shown (**a**); interestingly, near the Saraceno river, debris movements are detected. SKATER optimal clustering is shown in panel (**b**); the spatial distribution of the LOS velocities retrieved by the SPINUA algorithm is shown in panel (**c**).

In the Trebisacce-Villapiana coastal plain, both hotspots and coldspots were detected. For example, particular mentions deserve the subsidence (coldspot) area inherent in the Villapiana shore and the uplifting (hotspot) area of Trebisacce. A magnified view of these areas is presented in Figure 9.

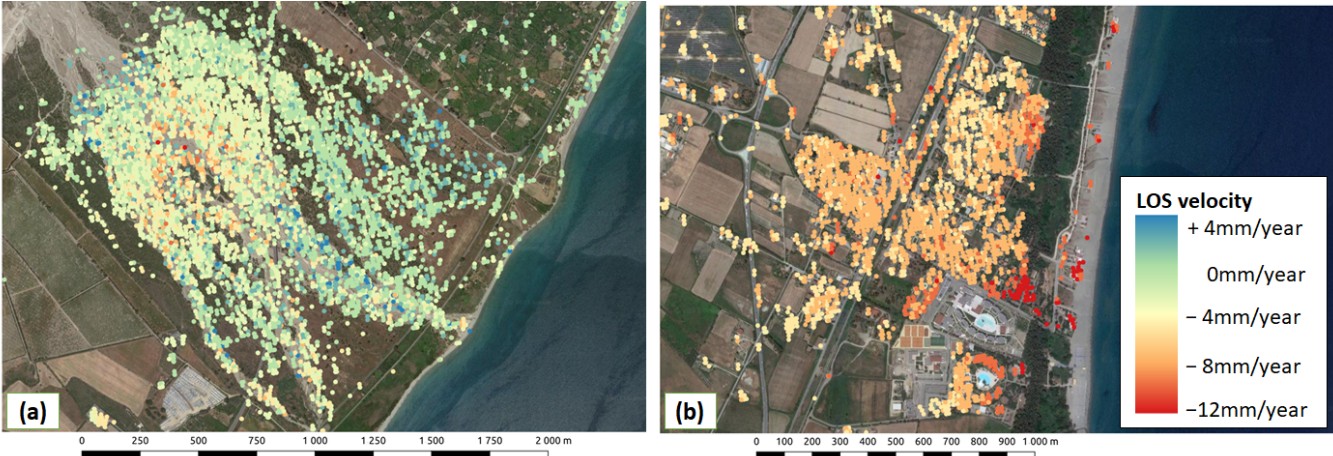

**Figure 9.** Two areas of interest in the Trebisacce-Villapiana region: the mouth of river Saraceno near Trebisacce (**a**) and the Villapiana shore subsidence (**b**).

The figure shows two elements of interest. The mouth of the river Saraceno near Trebisacce-Villapiana. The river shows the presence of extremely heterogeneous LOS velocities, ranging from −10 mm to 10 mm per year, probably corresponding to superficial debris movements. Trebisacce shows a relevant uplifting movement along the LOS. Finally, for what concerns the shore of Villapiana, a significant subsidence (3 mm per year) is detected. Maximum values of around 7 ∼ 13 mm per year are also observed.

### 4.4. Policoro

The Policoro coastal plain was considered as a unique cluster because the BSS/TSS ratio examination suggested that the clusterization was not reliable in this case. Then, LISA analysis was performed over the whole region; even in this case, some hotspots and coldpots were detected. Some particular uplifting areas were found along the Cardonna, Canna and San Nicola torrents; interestingly, portions of the SS 106 Jonica highway were both affected by hotspots and coldspots: results are shown in Figure 10.

In particular, two elements of interest deserve further investigation: the SS 106 Jonica highway bridge near Nova Siri Scalo beach and Policoro Lido shore, see Figure 11.

In particular, along this bridge, extremely heterogeneous LOS velocities were detected; these regions, outlined in yellow dashed circles, showed velocities ranging from −15 mm to 15 per year. More specifically, this phenomenon occurs in proximity of a bridge. Policolouro Lido showed a small but relevant subsidence hotspot with LOS velocities of about −16 mm per year. Maximum values of velocities along the LOS (∼13 mm per year) were observed. Moreover, coastal subsidence was also observed along the shorelines ([−6, −10] mm/year).

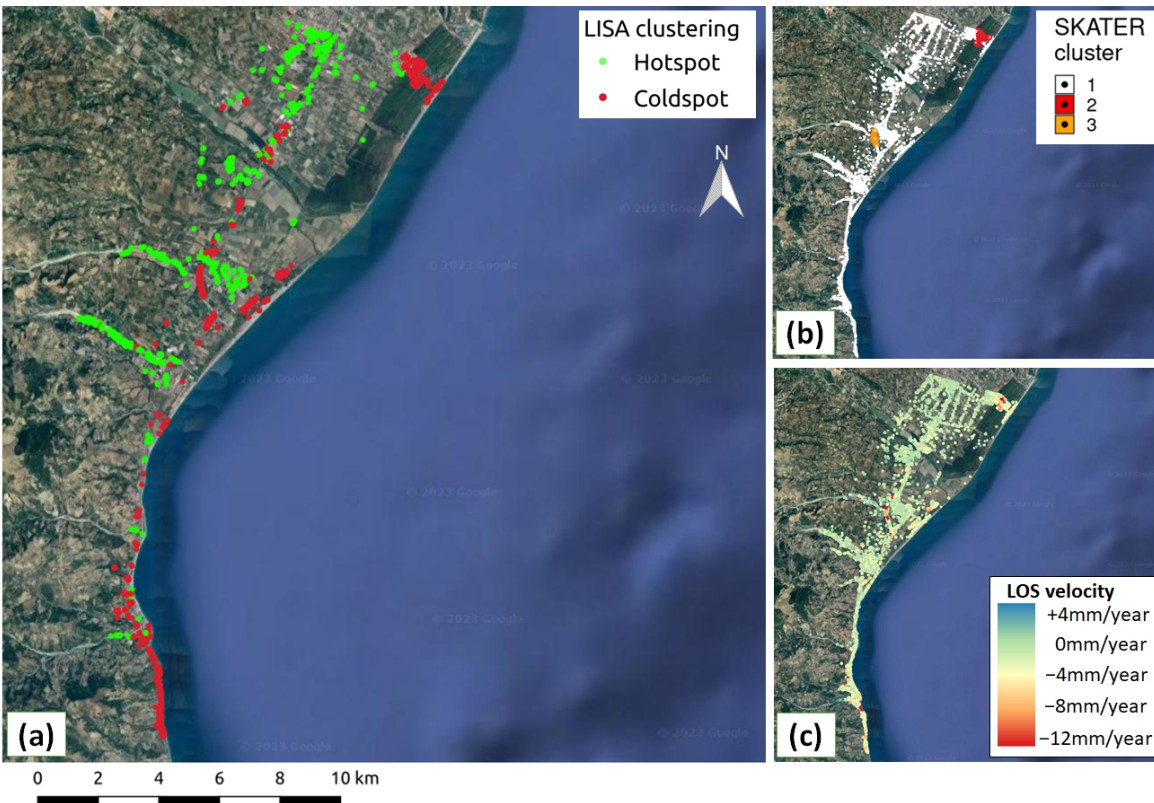

**Figure 10.** LISA analyses of Policoro dataset: coldspots (red) and hotspots (green) are shown (**a**); the analysis reveals three major areas of concerns, namely, two portions of the SS 106 Jonica highway and a subsidence coldspot in Policoro Lido. SKATER optimal clustering is shown in panel (**b**); the spatial distribution of the LOS velocities retrieved by the SPINUA algorithm is shown in panel (**c**).

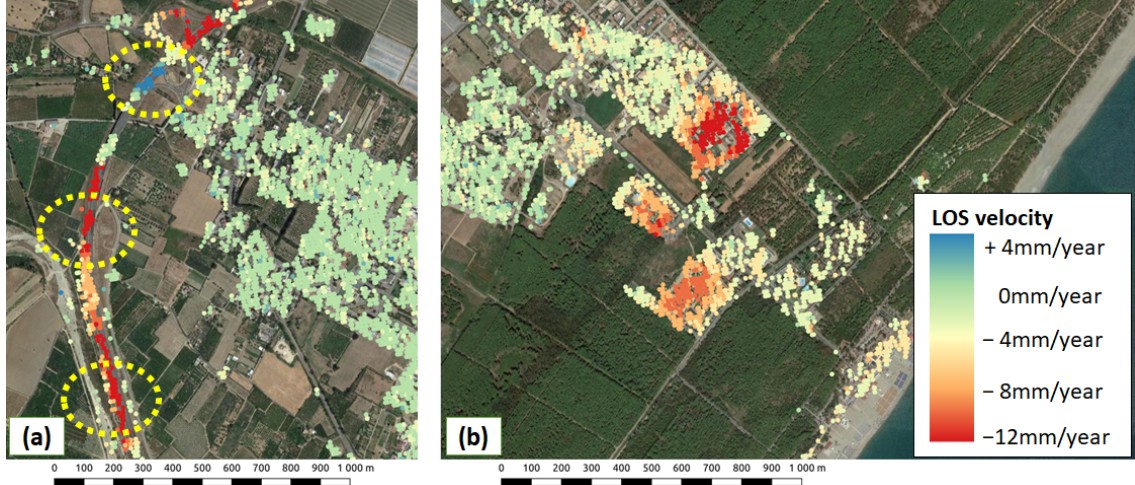

**Figure 11.** Subsidence phenomena in the Policoro area: the SS 106 Jonica highway (**a**) and the Policoro Lido settlement (**b**). For what concerns the highway, traits with extremely varying velocities, ranging from −15 mm to 15 per year are highlighted (dotted circles).

## 5. Discussion

Here, we presented a novel workflow that combines a powerful and computationally efficient clustering algorithm such as SKATER with a local analysis outlining homogeneous patterns characterized by lesser scales than SKATER clusters or local anomalies. The main feature offered by SKATER is that it is a spatially constrained algorithm, a decisive

feature when dealing with geographical analyses. An immediate consequence is that SKATER clusters do not yield extremely parcelled segmentations but tend to cover more extended areas.

For example, in the Sibari region, only three clusters were detected; one including the majority of points characterized by stable LOS velocities, the other two clusters characterized by subsidence. Analogously, three clusters were found by SKATER in the Trebisacce-Villapiana area; one for subsidence in the south, one uplifting region in the north, and a stable region in the middle. Finally, according to SKATER, the whole Policoro area was considered as a unique homogeneous cluster. However, it is reasonable to assume that by further inspection, a more detailed characterization of local phenomena could arise. This is where LISA analyses become useful.

In fact, LISA analysis allows us to distinguish within the Sibari region some specific subsidence areas which would have been grouped together if considering only the SKATER results. In particular, our findings outlined the subsidence affecting the Sibari lakes surroundings, which is particularly interesting if considering the residential areas in the surroundings and the fact that it is located 2.5 m above the sea level. Additionally, the subsidence affecting Corigliano Calabro was highlighted: on the one hand, we found a subsidence induced by anthropic pressure in the industrial area, probably related to the continuous water supply for industrial needs affecting the water well beneath; on the other hand, the analyses revealed the subsidence of Salicetti, a coastal fraction of Corigliano Calabro. In fact, subsidence of coastal regions, such as that of Salicetti or the Sibari port (another point of interest for subsidence) should be carefully monitored, especially considering the combined action of subsidence and sea-level increment due to climate change.

Further details were provided by LISA for Trebisacce-Villapiana, too. The mouth of the Saraceno river showed an interesting behavior with extremely heterogeneous LOS velocities; it is reasonable to assume this is due to debris, moreover it is a region far from inhabited areas, nevertheless these movements need to be monitored. The uplifting movement of Trebisacce is already known [57]; this can be considered an indirect validation of the robustness of these findings.

Analogous considerations arise looking at the Policoro region where a general coastal subsidence was observed. Again, this finding is confirmed by previous studies [58,59], thus validating the proposed procedure. This general subsidence is expected to involve a coastal loss of 1 m per year, hence suggesting a continuous monitoring. Additionally, the SS106 Jonica highway deserves a particular mention; specifically, the trait near Nova Siri (lat 40.135, lon 16.625) showed LOS velocities ranging from $-15$ mm to 15 mm per year. Finally, a significant subsidence ($-16$ mm per year) cluster was observed within the Policoro Lido fraction. To the best of our knowledge, this phenomenon has not been previously observed and deserves further investigations.

It is worth mentioning that LISA analyses also revealed local anomalies; less than 1% of examined PSs consisted of isolated points. In these cases, we chose to neglect such anomalies because we were unable to ensure their statistical robustness or to verify with on ground observations if they were related to interesting phenomena. Accordingly, future work could refine the proposed approach. Nevertheless, the presented findings suggest unanimously that this pipeline can be suitably adopted for environmental and infrastructural monitoring.

## 6. Conclusions

In this work, we presented a novel workflow for PSI analyses; specifically, we adopted SKATER and LISA methods to perform spatially constrained clusterization and a subsequent investigation of local patterns or anomalies. We demonstrated how SKATER clustering represents a suitable tool for PSI in that the clusters it yields are a faithful representation of the ground deformations returned by PSI when performing regional-scale analyses. Nevertheless, the large clusters returned by SKATER include local patterns that, without the subsequent LISA analysis, would be inevitably missed. In particular,

we showed the presence of significant local subsidence and uplifting phenomena in the examined regions. These phenomena being due to anthropic pressure such as industrial or touristic areas, as well as being due to natural causes, it is of paramount importance to have accurate tools with which to monitor them. This is of particular interest for both environmental and infrastructural monitoring. To this aim, it is also worth mentioning that the National Recovery and Resilience Plan presented by Italy, as part of the the Next Generation EU programme, has explicitly allocated huge resources for computing infrastructures deputed to environmental monitoring; hence, the development of novel strategies and approaches which exploit the massive informative content provided by Earth observation is not only useful but encouraged.

**Author Contributions:** Conceptualization, N.A. and R.C.; methodology, N.A. and R.C.; software, R.C.; formal analysis, R.C.; writing—original draft preparation, N.A. and R.C.; writing—review and editing, all the authors; visualization, N.A. and R.C.; supervision, N.A. and R.B.; funding acquisition, R.B. All authors have read and agreed to the published version of the manuscript.

**Funding:** Project funded under the National Recovery and Resilience Pan (NRRP), Mission 4 Component 2 Investment 1.4—Call for tender No. 3138 of 16 December 2021 of Italian Ministry of University and Research funded by the European Union—NextGenerationEU. Award Number: Project code: CN00000013, Concession Decree No. 1031 of 17 February 2022 adopted by the Italian Ministry of University and Research, CUP H93C22000450007, Project title: "National Centre for HPC, Big Data and Quantum Computing.

**Data Availability Statement:** Data used in this paper are open source, they can be download from the official site https://sentinel.esa.int/web/sentinel/sentinel-data-access (accessed on 8 May 2023) without any charges. More information about the SPINUA processing or the code for SKATER and LISA analysis is available on request.

**Acknowledgments:** Authors would like to thank IT resources made available by ReCaS, a project funded by the MIUR (Italian Ministry for Education, University and Research) in the "PON Ricerca e Competitività 2007–2013-Azione I-Interventi di rafforzamento strutturale" PONa3_00052, Avviso 254/Ric, University of Bari. This paper has been supported by the TEBAKA (TErritorial BAsic Knowledge Acquisition project "Avviso MIUR n. 1735 del 13/07/2017".

**Conflicts of Interest:** The authors declare no conflict of interest.

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
