# Peer review of "PSI Spatially Constrained Clustering: The Sibari and Metaponto Coastal Plains"

_remotesensing, doi:10.3390/rs15102560_

Round 1

Reviewer 1 Report

The scope and methodology of work are acceptable. However, the presentation of the results is poor. I suggest the authors provide better-quality images so that the resolution and clarity of the content would be good.

Reviewer 2 Report

The paper “PSI spatially constrained clustering: the Sibari and Metaponto costal plains”, reports an interesting research work about the evaluation of on-ground displacements through the application of innovative techniques. In particular, the approach proposed by the authors is applied to the Sibari and Metaponto coastal plains. In general, the paper is well-organized in its different Sections and the methods used are clearly described in the text. The results obtained are clearly commented in Section 5. Only minor improvements/corrections are suggested before the publication in Remote Sensing Journal:

-        a significant improvement of the quality of the Figures is recommended;

-        line 13: standardize the keywords;

-        Table 1: rewrite according to the journal guidelines;

-  Section 6: add some considerations about the limitations of the results obtained and highlight the original aspects of the work.

Reviewer 3 Report

The author of the manuscript proposed a new computational framework for monitoring ground displacement. Although there have been many studies in this area, the study of this manuscript is still valuable. There are also some problems in the manuscript that can be accepted after revising, as follows:

(1) There are a large number of abbreviations in the manuscript that are not explained, which can make it difficult for readers to understand. For example, in the abstract, some are written with explanations, while others are not. Type situations also exist in the text.

(2) The quality of all figures in the manuscript is low, and it is not possible to clearly read the illustrations and text in the drawings. Please improve the quality of the figure representation to make the manuscript more readable.

(3) How can the correctness of the calculation method be verified? Please describe the validation issues of the calculation method in more detail to let the reader know that the method is indeed reliable.

Reviewer 4 Report

The work submitted to me for evaluation on the use of psi to determine land subsidence is very interesting and has publication potential, however, it requires necessary major corrections.

First, the authors should be more clear about what is new about the presented paper, what this paper brings to the knowledge on the subject. Secondly, the quality of the drawings left much to be desired. The drawings are mostly of poor quality and, above all, illegible. All should be corrected. The literature review should also be supplemented: for example, with other works on land subsidence in industrial areas, e.g.:

Solarski, M., Machowski, R., Rzetala, M. et al. Hypsometric changes in urban areas resulting from multiple years of mining activity. Sci Rep 12, 2982 (2022). 

Nádudvari, Ádám. "Using radar interferometry and SBAS technique to detect surface subsidence relating to coal mining in Upper Silesia from 1993-2000 and 2003-2010" Environmental & Socio-economic Studies, vol.4, no.1, 2016, pp.24-34. https://doi.org/10.1515/environ-2016-0003

Round 2

Reviewer 3 Report

Most of the modifications to this manuscript have met the requirements, but there are still some details that need to be revised. For example, when multiple figures are placed together for comparison, each figure needs to be annotated and explained.

Author Response

We thank reviewer 3 for his significant contribution to the overall quality of our manuscript. As requested, we have annotated the plots and the maps when merged together in a single figure. 

Reviewer 4 Report

The article has been corrected according to my suggestions, so I believe it can be published in its current form.

Author Response

We thank reviewer 4 for his significant contribution to the overall quality of our manuscript.